# Functional Analysis of *Malus halliana* *WRKY69* Transcription Factor (TF) Under Iron (Fe) Deficiency Stress

**DOI:** 10.3390/cimb47070576

**Published:** 2025-07-21

**Authors:** Hongjia Luo, Wenqing Liu, Xiaoya Wang, Yanxiu Wang

**Affiliations:** 1College of Horticulture, Gansu Agricultural University, Lanzhou 730070, China; jia.jialuo@163.com (H.L.); 15720149915@163.com (W.L.); 18095526010@163.com (X.W.); 2Forestry Technique Extension Station of Gansu Province, Lanzhou 730070, China

**Keywords:** *WRKY69* gene, genetic transformation, bioinformatics analysis, Fe deficiency stress

## Abstract

Fe deficiency in apple trees can lead to leaf chlorosis and impede root development, resulting in significant alterations in signaling, metabolism, and genetic functions, which severely restricts fruit yield and quality. It is well established that WRKY transcription factors (TFs) are of vital significance in mediating plant responses to abiotic stress. Real-time quantitative fluorescence (RT-qPCR) analysis displayed that Fe deficiency stress can significantly induce *WRKY69* TF gene expression. However, the potential mechanisms by which the *WRKY69* gene involved in Fe deficiency stress remains to be investigated. To address this limitations, the *WRKY69* gene (MD09G1235100) was successfully isolated from apple rootstock *Malus halliana* and performed both homologous and heterologous expression analyses in apple calli and tobacco to elucidate its functional role in response to Fe deficiency stress. The findings indicated that transgenic tobacco plants exhibited enhanced growth vigor and reduced chlorosis when subjected to Fe deficiency stress compared to the wild type (WT). Additionally, the apple calli that were overexpressed *WRKY69* also exhibited superior growth and quality. Furthermore, the overexpression of the *WRKY69* gene enhanced the ability of tobacco to Fe deficiency stress tolerance by stimulating the synthesis of photosynthetic pigments, increasing antioxidant enzyme activity, and facilitating Fe reduction. Additionally, it increased the resistance of apple calli to Fe deficiency stress by enhancing Fe reduction and elevating the activity of antioxidant enzymes. For example, under Fe deficiency stress, the proline (Pro) contents of the overexpression lines (OE-2, OE-5, OE-6) were 26.18 mg·g^−1^, 26.13 mg·g^−1^, and 26.27 mg·g^−1^, respectively, which were 16.98%, 16.76%, and 17.38% higher than the proline content of 22.38 mg·g^−1^ in the wild-type lines, respectively. To summarize, a functional analysis of tobacco plants and apple calli displayed that WRKY69 TF serves as a positive regulator under Fe deficiency stress, which provides candidate genetic resources for cultivating apple rootstocks or varieties with strong stress (Fe deficiency) resistance.

## 1. Introduction

Iron (Fe) micronutrient displays a pivotal role in the growth, development, and reproduction of plants [1]. It is particularly important as a key component in respiration, chlorophyll (Chl) synthesis, and the process of photosynthesis [2,3,4,5,6,7,8]. Fe, as a component of certain essential oxidoreductases, serves as a significant electron carrier in the physiological reactions of plants [9,10]. Despite the high levels of Fe found in soils, its availability frequently remains limited due to the formation of insoluble Fe(III)-hydroxide complexes in calcareous soils [11,12,13]. These complexes pose a major challenge to the absorption and utilization of Fe by plants and often lead to Fe deficiency, which manifests as leaf chlorosis. This condition hinders plant growth and development, severely limiting crop yield and quality [14]. It was discovered that there is an increase in the levels of hydrogen peroxide and reactive oxygen species when experiencing iron deficiency in plants, along with alterations in the activity of certain antioxidant enzymes [15]. This suggests that an environment lacking sufficient iron might induce modifications in the plants’ antioxidant systems and lead to the development of self-protective mechanisms [16].

Transcription factors (TFs) play a crucial role in the development of self-defensive mechanisms in plants, enabling them to cope with the challenges posed by an Fe-deficient environment [17,18]. Among these, transcription factors, such as *WRKY* TF, represent a significant category within the plant TF family, as they modulate gene expression levels downstream by specifically regulating target genes, which is capable of swiftly responding to various stresses [19,20,21,22]. In the case of plants, the expression levels of this TF are not consistently elevated, as they are influenced by the surrounding environment. Notably, when plants encounter external abiotic stress, the expression of the WRKY transcription factor can increase rapidly [23]. When aluminum treatment is applied to *Arabidopsis thaliana*, the gene *AtWRKY46* inhibits the expression of the crucial aluminum-tolerant gene ALUMINUM-ACTIVATED MALATE TRANSPORTER 1 (*ALMT1*) by directly binding to the W-box within its promoter region. This interaction negatively impacts the plant’s response to aluminum toxicity [24]. In contrast, *AtWRKY46* can rapidly respond to iron deficiency stress by upregulating the iron transporter gene *VITL1*. This process facilitates the transfer of iron from the roots to the aerial parts of the plant, thereby meeting the iron requirements of the aboveground tissues [25]. Moreover, WRKY transcription factors play a crucial role in regulating plant responses to temperature stress. In comparison to wild-type Arabidopsis, the overexpression of *CsWRKY46* in Arabidopsis resulted in elevated proline levels, along with reduced electrolyte permeability and malondialdehyde levels when exposed to low-temperature conditions. Furthermore, there was an increase in the expression of downstream genes associated with low-temperature stress, specifically *RD29A* and *COR47*, which contributed to the enhanced cold resistance of the transgenic Arabidopsis [26].

The Loess Plateau in Northwest China is the main producing area of apple cultivation in China, but salinization often occurs in this area, which leads to the decrease in iron solubility and leads to the deficiency of iron nutrition, which seriously affects the quality and yield of fruit. When iron deficiency occurs, plants can promote Fe absorption, distribution, and utilization by changing the expression levels of multiple genes or tf, so as to combat Fe deficiency adversity. In apple, a new WRKY gene, *MxWRKY53*, was isolated from *Malus xiaojinensis* and introduced into Arabidopsis, which significantly enhanced the iron tolerance of Arabidopsis [27]. However, only a limited number of studies have focused on the role of the WRKY gene in relation to Fe deficiency in apple. Previously, transcriptomic analysis revealed that WRKY69 was a differentially expressed TF under Fe deficiency conditions. This finding was further validated by RT-qPCR, which confirmed that WRKY69 expression was significantly upregulated in response to Fe deficiency. Consequently, this TF was cloned for subsequent genetic transformation and functional characterization. Therefore, the study provides a theoretical foundation for identifying Fe deficiency-resistant apple varieties.

## 2. Materials and Methods

### 2.1. Plant Materials and Treatments

For the analysis of gene expression, tissue culture seedlings of the apple rootstock *Malus halliana* (diplont, a rootstock of apples native to Gansu Province) were selected for root culture after being subcultured for 30 days in a root medium containing 0.5× macronutrients, 0.5× micronutrients, 0.4 mg/L NAA, 7 g/L agar, and pH adjusted to 5.8–6.0. The seedlings were stored in Room 601, College of Horticulture and Gardening, Gansu Agricultural University, Lanzhou City, Gansu Province. After the seedlings took root, they were inserted into 50 mL of MS liquid medium without Fe for treatment and fixed by the paper bridge method [28]. Subsequently, Fe deficiency stress treatment was initiated, and leaf samples were collected at 0, 6, 12, 24, 48, and 72 h post-treatment. The concentration of Fe deficiency was determined according to the guidelines proposed by Han (1994) [29], setting the level at 0 μM Fe (—Fe). During the sampling process, each time point included three biological replicates, with five seedlings in each replicate.

Tobacco (*Nicotiana tabacum*) culture was substituted on MS medium every 30 days for 16 and 8 h of light and dark cycles, respectively, at 25 °C. Conversely, ‘Wanglin’ apple calli was induced by Japanese scholars with ‘Wanglin’ apple embryo and was introduced into China by Professor Hao Yujin from Shandong Agricultural University during his post-doctoral study and authorized to use in our laboratory. Additionally, they were subcultured every 20 days on MS medium supplemented with 1.5 mg/L 2,4-D and 0.4 mg/L 6-BA and maintained in complete darkness at a constant temperature of 25 °C.

### 2.2. Bioinformatic Analysis of WRKY69 Gene

The protein sequences of the *WRKY69* gene were obtained from apple genome database (https://phytozome-next.jgi.doe.gov/, accessed on 21 September 2024). Subsequently, the basic physicochemical properties of these proteins were predicted using ExPASy (https://web.expasy.org/protparam, accessed on 21 September 2024). DNAMAN software (v9.0) was used for protein sequence analysis of this gene in different species (accessed on 21 September 2024). Software MEGAX (v11.0.13) was used to construct a neighborhood (NJ)-based phylogenetic tree (accessed on 22 September 2024).

### 2.3. Cloning and Expression Vector Construction of WRKY69 Gene

*Malus halliana* seedlings were used as materials for RNA extraction, with an RNA extraction kit provided by BioTeke Corporation, based in Beijing, China. The PrimeScript™ RT Reagent Kit with gDNA Eraser (Perfect Real Time) from TaKaRa (Kusatsu, Japan) was employed to conduct reverse transcription. Specific primers (Appendix A) were designed for amplification through qRT-PCR, following a search of the apple genome database for the *WRKY69 * CDS sequence, utilizing DNAMAN software (v9.0). The qRT-PCR template was derived from the cDNA of *M. domestica* plantlets. Quantitative data analysis was conducted using the 2^−ΔΔCT^ method [30]. The PCR procedures included an initial step at 95 °C for 5 min, followed by 30 s at 95 °C, 42 s at 57 °C, 100 s at 72 °C, and a final extension of 10 min at 72 °C.

The PCR products were subjected to electrophoresis on an agarose gel to isolate the target genes. Subsequently, the pMD19-T vector was applicated to ligated into the products. After obtaining the *WRKY69* plasmid, on the one hand, homologous arms were added to the primers, and the plasmid was used as a template for gene expansion, purification, and recovery. On the other hand, the pRI101 vector was double-digested with two enzymes *(Sma I* and *Kpn I*) for linearization treatment and then ligated with the recombinant enzyme Exnase II (C112, purchased from Vazyme, Nanjing, China), transferred to *Escherichia coli*, and sent to Shanghai Shengong Co., LTD. (Shanghai, China), for sequencing. If the sequencing is correct, the plasmid is extracted; that is, the gene is successfully connected to the PRI101 expression vector. Subsequently, *Agrobacterium tumefaciens* GV3101 was transferred by the freeze–thaw alternating method to prepare for the later genetic transformation [31].

### 2.4. Agrobacterium-Mediated Transformation of Tobacco and Apple Calli

Following the approach established by Xu et al. (2019) [32], albeit with minor variations, tobacco leaves were subjected to trauma and subsequently infected with Agrobacterium for 15 to 20 min. Subsequently, the leaves were subjected to a pre-culturing process on a medium free of antibiotics, maintained in dark conditions for a duration of 2 to 3 days. Next, they were transferred to a medium containing 250 mg/L cephalosporin and 30 mg/L kanamycin. Once the buds attained an approximate length of 1.5 cm, they were harvested and transferred to a rooting medium for further cultivation. The DNA from the regenerated seedlings was extracted and analyzed using PCR techniques.

In accordance with the approach outlined by Li et al. (2017) [33], apple calli from the suspension culture during the exponential growth stage underwent subculturing for roughly two weeks. The calli was immersed in a solution of Agrobacterium for a duration of 15 to 20 min, after which the solution was removed. Subsequently, the calli was placed on an antibiotic-free solid medium and incubated in darkness for 48 h. Following this incubation, the calli was rinsed with sterile water 3 to 5 times, with each rinse lasting three minutes. The calli was then transferred to a medium supplemented with 250 mg/L of cephalosporin and 30 mg/L of kanamycin, facilitating the stable growth of the resistant callus over a period of 30 to 60 days. Subsequently, RNA from apple calli was extracted to quantitatively detect WRKY69 TF expression by real-time fluorescence (qRT-PCR).

### 2.5. Treatment with Fe Deffciency in Transgenic Tobacco and Apple Calli

Tobacco plants, both transgenic and wild-type, were cultivated on MS solid medium for a period of 20 days. Following this, they were transferred to Fe-sufficient (+Fe) and Fe-deficient (-Fe) liquid media for 30 days using the Filter Paper Bridge method. After this treatment, the phenotypes were assessed, and various related indices were measured. Concurrently, overexpressed and wild-type apple calli were also maintained on MS solid medium for 15 days, followed by culture in Fe-sufficient and Fe-deficient liquid media for 20 days to evaluate their phenotypic characteristics and determine the relevant indices. Fe was supplied in the form of Fe-EDTA.

### 2.6. Transgenic Tobacco Plants and Apple Calli Under Fe (Iron) Deficiency Stress and Determination of Related Indicators

In the DAB (3,3′-Diaminobenzidine) staining process, leaf samples were soaked in 50 mM DAB solution for 24 h and later decolorized in 95% ethanol until colorless. In the NBT (nitro blue tetrazolium) staining process, the samples were soaked in 50 mM NBT solution for 4 h and then decolorized in 95% ethanol until white in appearance. The determination of chlorophyll content followed the protocol of Cheng (2020) [34], and the analysis of proline content followed the method of Ferreira Júnior (2018) [35]. Malondialdehyde (MDA) content quantification was conducted by the method described by Xiao and Zhou (2023) [36]. The spectrophotometric determination of superoxide dismutase (SOD), peroxidase (POD), and catalase (CAT) activities was conducted using kits from Beijing Solarbio Science & Technology Co., Ltd., located in Beijing, China. The conductivity was assessed via the conductivity method (DDS-307) as outlined by Bajji et al. (2002) [37]. Each experimental trial was performed in triplicate. Finally, Ferric Chelate Reductase (FCR) activity was evaluated following the modified approach of Schikora and Schmidt (2001) [38]. The analysis of acidification was conducted based on the technique described by Zhao et al. (2016) [39].

### 2.7. Data Analysis

Statistical data was conducted using Excel 2019, and data analysis was performed with SPSS 21.0. Visual representations were created using Origin 2021. The significance of the test results was evaluated using the least significant difference (LSD) method within the ANOVA framework.

## 3. Results

### 3.1. Protein Sequence Analysis and Evolutionary Relationship Analysis of WRKY69 and Its Homologous Genes

As shown in Figure 1A, the evolutionary tree indicates that *WRKY69* and *WRKY65* present the closest kinship, followed by *WRKY11* and *WRKY17*. In addition, multiple sequence alignment revealed that the protein sequence of *WRKY69* shows certain differences from that of its homologous genes at both the N-terminal and C-terminal (Figure 1B).

The expression of *WRKY69* consistently increased compared to the control (0 h), peaking at 48 h with an expression level 7.156 times greater than that of the control (Figure 2). Furthermore, the homologous genes of *WRKY69*, *WRKY11*, *WRKY17*, and *WRKY65* were significantly upregulated under Fe deficiency stress, while *WRKY74* and *WRKY74-like* were significantly downregulated. However, the changes in their expression levels under the stress were not as significant as those of *WRKY69*. Therefore, the *WRKY69* transcription factor was selected for cloning for genetic transformation.

### 3.2. Analysis of the WRKY69 Gene

The *WRKY69* gene (MD09G1235100) was isolated from *Malus halliana*, with an open reading frame of 753 bp. This protein possesses a molecular weight of 61.739 kDa and a theoretical isoelectric point (pI) of 5.10, classifying the *WRKY69* gene as an acidic protein. Additionally, it possesses an aliphatic index of 28.29 and an instability index of 45.46, suggesting that it is a relatively unstable protein. The average hydrophilicity value is 0.916 (Table 1). These findings indicate that the *WRKY69* protein exhibits characteristics of instability and acidic hydrophobicity.

A comparison of the amino acid sequences of apple *WRKY69* proteins and their WRKY homologues from various species was conducted using the NCBI database in conjunction with DNAMAN software (Figure 3). It was observed that the C-terminal regions of *WRKY69* proteins, as well as those from other species, exhibited some degree of variation; however, they typically contained a C_2_H_2_ zinc finger motif. At the N-terminus, a highly conserved heptapeptide sequence, WRKYGQK, indicates the presence of the WRKY structural domain. This finding suggests that *WRKY69* proteins, along with the majority of their homologues, are classified within the second class of WRKY transcription factors. Using the MEGAX (v11.0.13) software to construct evolutionary trees, the results of the phylogenetic analysis indicate that *M. domestica* (XP_008381191.1) clusters with *Malus baccata* (TQE05538.1) and *Pyrus × bretschneideri* (XP_009365282.2) within the same subgroup. These findings suggest that the *WRKY69* gene in *M. domestica* is most closely related to those in *Malus baccata* and *Pyrus × bretschneideri*, implying that it may share analogous biological functions (Figure 4).

### 3.3. Identiffcation of Transgenic Tobacco and Overexpressed Apple Calli

Compared to wild-type plants, the transgenic tobacco and apple calli displayed significantly higher levels of *WRKY69* expression, which proved that overexpressed *WRKY69* transgenic tobacco and apple calli were obtained (Figure 5).

### 3.4. Resistance of Transgenic WRKY69 Tobacco to Fe Deffciency Stress

As illustrated in Figure 6, both wild-type and transgenic tobacco exhibited healthy growth under standard conditions, characterized by vibrant green foliage and no significant differences. However, the wild-type control displayed more pronounced leaf chlorosis and reduced growth when subjected to Fe deficiency stress compared to the transgenic variety. Various physiological parameters in tobacco were altered due to Fe deficiency stress (Figure 7). When Fe content was sufficient, peroxidase (POD) and superoxide dismutase (SOD) did not differ significantly between the three transgenic lines (OE-2, OE-5, and OE-6) compared to WT control, while catalase (CAT) showed some differences. However, under the conditions of Fe deficiency, the activities of the antioxidant enzymes (POD, CAT) in transgenic tobacco were markedly increased relative to the WT. These findings suggest that the overexpression of the *WRKY69* gene enhances the activity of the antioxidant enzyme system in tobacco. Under the conditions of Fe deficiency stress, the levels of FCR activity, soluble protein, and proline in the transgenic lines significantly surpassed those of the WT control, while malondialdehyde (MDA) and hydrogen peroxide (H_2_O_2_) levels were markedly reduced in comparison to the WT. Consequently, the capacity of plant cells to scavenge reactive oxygen species (ROS) improved, resulting in lower accumulations of superoxide (O_2_^−^) and H_2_O_2_. Furthermore, the levels of chlorophyll a, chlorophyll b, and chlorophyll a + b of the three transgenic strains were significantly different from those of WT under the condition of Fe sufficiency. Among them, the contents of these pigments in transgenic strain OE-2 were significantly higher than those in wild type. However, under the conditions of Fe deficiency, the content of these pigments in the three transgenic lines was significantly higher than that in the WT lines. This study demonstrated that the overexpression of the *WRKY69* gene could enhance the synthesis of photosynthetic pigments and increase chlorophyll contents in transgenic plants.

### 3.5. Functional Analysis of Apple Calli with Overexpression of WRKY69 Gene Under Iron Deficiency Stress

Figure 8 clearly illustrates that, under the condition of Fe sufficiency, the overexpressed apple calli exhibited no significant differences compared to the control. However, when subjected to iron-deficiency conditions, marked variations were observed between the transgenic lines (OE-2, OE-5 and OE-6) and the wild type, with the growth of the overexpressed apple calli being notably superior. Furthermore, under iron deficiency, the periphery of the overexpressed apple calli turned yellow due to bromocresol violet staining, indicating that the *WRKY69* transgenic lines released more H^+^ into the medium. This acidification of the medium resulted in the yellow coloration observed (Figure 9). In conclusion, when analyzed alongside WT calli, POD, SOD, and CAT activities of the lines OE-2, OE-5, and OE-6 significantly increased. In addition, they showed significant increases in soluble protein, proline content, and FCR activity, while MDA and H_2_O_2_ levels were significantly lower than those of the control group, especially under iron deficiency conditions (Figure 10).

## 4. Discussion

### 4.1. Bioinformatics Analysis of WRKY69 TF

Growth and development in plants can be influenced by a variety of abiotic factors, including drought conditions in the soil, saline–alkali stress, and nutrient imbalances [40,41]. Fe deficiency is prevalent in plants globally and leads to various issues, such as the inability to synthesize chlorophyll, damage to organelles, and leaf chlorosis [42,43]. Furthermore, it disrupts photosynthesis [44,45], impacts plant growth and development, and significantly limits both the quality and quantity of fruit yield [46,47].

The WRKY TF family is prevalent across various plant species. In recent years, studies have focused on its presence in plants, such as soybean [48], rice [49], maize [50], and apple [51]. These plants typically possess one to two conserved WRKY structural domains, alongside a zinc finger motif in their amino acid sequences, categorized as C_2_H_2_-type and C_2_HC-type [52]. The presence of these two structures influences WRKY’s ability to effectively bind to the W-box (C/T)TGAC(T/C) [53,54]. Findings indicate that WRKY plays a crucial role in various physiological and biochemical regulatory mechanisms, particularly in how plants respond to both biotic and abiotic stresses [55,56]. Its primary emphasis is on developing resistance to salt and cold stress in graminaceous species. For instance, in tobacco, the overexpression of *MbWRKY4* has been shown to enhance the plant’s capacity to withstand salt stress [57], while *AtWRKY46* is involved in the plant’s response to iron deficiency stress [58]. Nevertheless, there is a limited number of studies regarding WRKY’s capability to withstand iron deficiency stress in fruit trees.

In this study, we extracted the *WRKY69* gene from *Malus halliana*. The fundamental physicochemical characteristics indicate that *WRKY69* is an unstable, acidic, and hydrophobic protein. Additionally, we utilized DNAMAN to perform multiple sequence comparisons between *WRKY69* and closely related amino acid sequences from various species, developed an evolutionary tree for homology assessment, and made initial predictions regarding the function of the target gene. The amino acid sequence of *WRKY69* exhibited the highest homology with *Malus baccata* and *Pyrus×bretschneideri*, suggesting a close phylogenetic relationship [59]. Concurrently, it was discovered that all of these WRKY transcription factors possess a heptapeptide sequence known as WRKYGQK, along with a C_2_H_2_-type zinc finger configuration. This suggests that *WRKY69* and its related sequences are categorized as class II WRKY transcription factors. Numerous WRKY transcription factors belong to class II, including *MbWRKY3* [60] and *PgWRKY2* [61], among others. Typically, these factors contribute to the synthesis of various substances, seed germination, and the ability to withstand multiple abiotic stresses, such as high temperatures, drought, cold damage, and salinity. For instance, the soybean genes *GmWRKY16* and *GmWRKY12*, classified as class II, can enhance plant resistance to drought and salt stress [62,63]. Our research also revealed that the homologous genes of *WRKY69* are primarily associated with abiotic stresses, including low temperatures and high salinity. Therefore, the target genes we investigated are likely implicated in the mechanisms through which plants respond to stress.

### 4.2. Overexpression of WRKY69-Enhanced Tolerance to Fe Deficiency in Transgenic Plants and Apple Calli

In this study, we produced *WRKY69* transgenic tobacco plants and apple calli and examined their phenotypes alongside various physiological parameters under the conditions of iron deficiency stress. When subjected to iron deficiency stress, the transgenic tobacco exhibited enhanced growth vigor and elevated chlorophyll levels; specifically, the contents of chlorophyll a (Chla), chlorophyll b (Chlb), and total chlorophyll (Chla + b) were significantly increased compared to the wild type (WT). For example, the Chlb levels in transgenic tobacco lines OE-2, OE-5, and OE-6 increased by 71.76%, 43.53%, and 29.52%, respectively, relative to WT, indicating that these transgenic plants improved chlorophyll production and strengthened their resistance to iron deficiency stress [64]. Simultaneously, the analysis of apple calli acidification demonstrated that apple calli tissues with overexpression exhibited enhanced proton (H^+^) pumping. This observation indicates that the overexpression in apple calli tissues elevates the activity of plasma membrane (PM) H^+^-ATPase, resulting in a reduced pH that increases the solubility of iron [65]. These findings suggest that the *WRKY69* gene act as positive regulators of plant response to Fe deficiency stress.

Research indicates that the H^+^-ATPase (AHA) AHA2, located on the root membrane of plants experiencing iron deficiency, releases protons, which subsequently reduces the pH level in the root zone [66]. Concurrently, there is an increase in the levels of H_2_O_2_ and reactive oxygen species (ROS) within the plant, accompanied by corresponding changes in the activity of specific antioxidant enzymes. This suggests that iron deficiency may alter the plant’s antioxidant system, leading to the development of a self-protective mechanism [66]. The activity of antioxidant enzymes in transgenic tobacco plants and apple calli was assessed, revealing that under conditions of iron deficiency, the levels of POD, SOD, and CAT enzymes exhibited a significant increase in transgenic apple calli, while transgenic tobacco plants showed significant higher POD and CAT activities when compared to WT. Furthermore, the content of H_2_O_2_ was markedly reduced by 14.659%–25.688%. This indicates that the antioxidant capacity of transgenic plants is enhanced through the reduction in reactive oxygen species (ROS), resulting in decreased membrane damage [15] and an increase in the activity of antioxidant enzymes [67,68]. These findings are consistent with those of Fang (2021) [69], who observed that the overexpression of *MsGSTU8* in genetically modified tobacco results in lower levels of ROS and malondialdehyde, while simultaneously enhancing the activities of antioxidant enzymes, thereby improving the plant’s resilience to iron deficiency stress [69].

Conversely, we assessed the activities of proline, malondialdehyde (MDA), soluble protein, and Ferric Chelate Reductase (FCR) in transgenic tobacco and apple calli. Under the conditions of iron deficiency, the levels of proline and soluble protein in the transgenic lines significantly surpassed those of the wild type (WT), indicating that transgenic plants possess enhanced abilities to modulate osmoregulatory compound levels, thereby mitigating oxidative stress [70]. Notably, MDA levels were significantly lower in overexpressing plants compared to wild-type plants, suggesting that while free radicals adversely affected all plants during iron deficiency stress, the extent of damage in transgenic plants was markedly less severe [71]. The findings of Wang (2018) [72] support these results, demonstrating that the overexpression of *MsWRKY11* leads to increased proline levels, reduced reactive oxygen species (ROS) levels, and enhanced salt tolerance in soybeans [72]. Furthermore, FCR activity is closely associated with the iron levels. A deficiency in environmental iron content results in a significant increase in FCR activity on the cytoplasmic membrane, prompting H^+^-ATPase to release protons, which subsequently lowers the pH level in the root zone [73]. In this research, the activity of FCR in transgenic tobacco plants and apple calli subjected to iron deficiency was found to be significantly greater than that observed in wild-type plants. Specifically, the OE-2, OE-5, and OE-6 lines of transgenic tobacco exhibited increases of 27.155%, 28.707%, and 26.767%, respectively, compared to WT. Conversely, in apple calli, the increases were noted at 24.590%, 24.836%, and 23.066%, respectively. These findings are consistent with the results presented by [74], suggesting that transgenic plants enhance the absorption of Fe^2+^ through the upregulation of FCR activity. Furthermore, the findings of this study indicate that *WRKY69* enhances resistance to Fe deficiency in transgenic tobacco plants and apple calli by stimulating the mechanisms for clearing reactive oxygen species (ROS).

## 5. Conclusions

We have confirmed that *WRKY69* genes play a crucial role in dealing with Fe deficiency stress in both tobacco and apple calli. In tobacco, the overexpression of the gene facilitates the production of photosynthetic pigments, increases the activity of antioxidant enzymes, and improves the efficiency of Fe^3+^ transformation, ultimately leading to enhanced tolerance to Fe deficiency. Similarly, Fe deficiency resistance of apple calli was boosted by increasing antioxidant enzyme activity, improving iron reduction, and heightening proton efflux, which contributes to a reduction in pH levels. In conclusion, the screening and identification of WRKY family members in apple that respond to Fe deficiency stress not only enrich the network of WRKY family members responding to Fe deficiency stress but also contribute to the innovation of apple stress-resistant germplasm resources, which holds significant scientific significance and application prospects. In the future, a variety of biochemical techniques will be utilized, such as electrophoretic mobility assay (EMSA), to search for the downstream target genes of the *WRKY69* TF, and experiments such as yeast double heterozygotic (Y_2_H) and co-immunoprecipitation (Co-IP) will be used to search for the interacting proteins of this transcription factor, which reveal the molecular mechanism by which the TF regulates Fe deficiency stress.

## Figures and Tables

**Figure 1 cimb-47-00576-f001:**
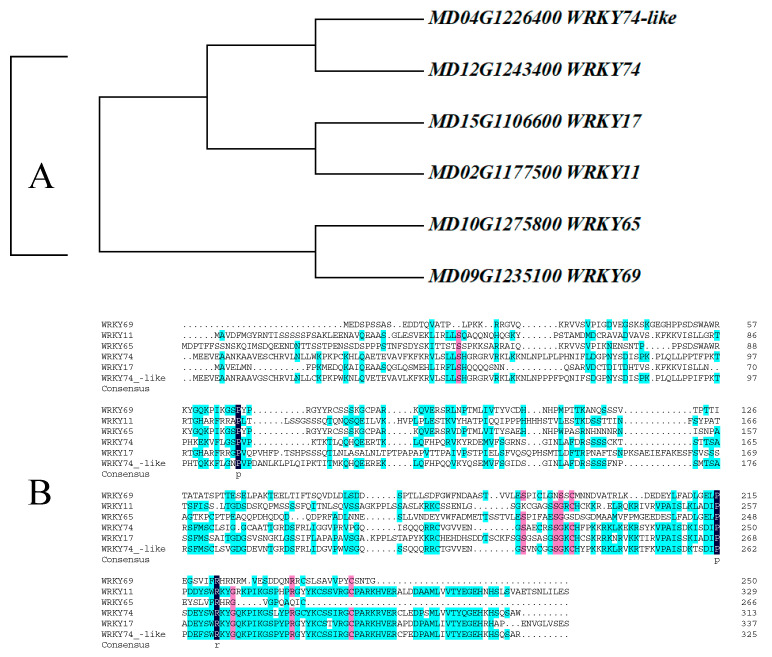
Protein sequence analysis and evolutionary relationship analysis of WRKY69 and its homologous genes in apple. (**A**). Phylogenetic analysis (**B**). Protein sequence analysis.

**Figure 2 cimb-47-00576-f002:**
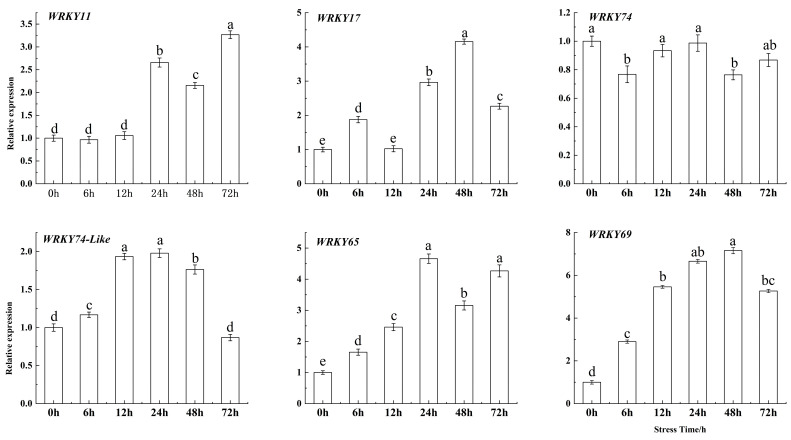
Expression levels of the *WRKY69* and its homologous gene in *M. domestica* seedlings were measured under Fe deficiency stress at 0, 6, 12, 24, 48, and 72 h. Note: error bars denote the SD of three replicates. Different letters above the bars indicated significant differences (*p* < 0.05) as assessed by one-way ANOVA and the least significant difference (LSD) test. The same below.

**Figure 3 cimb-47-00576-f003:**
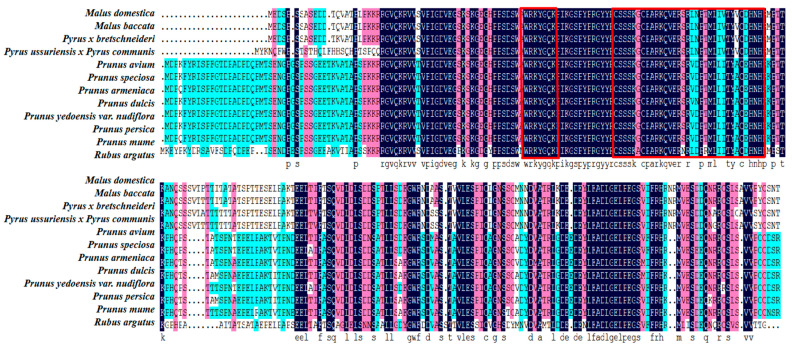
Protein sequence analysis of *WRKY69* gene and the protein of other species.

**Figure 4 cimb-47-00576-f004:**
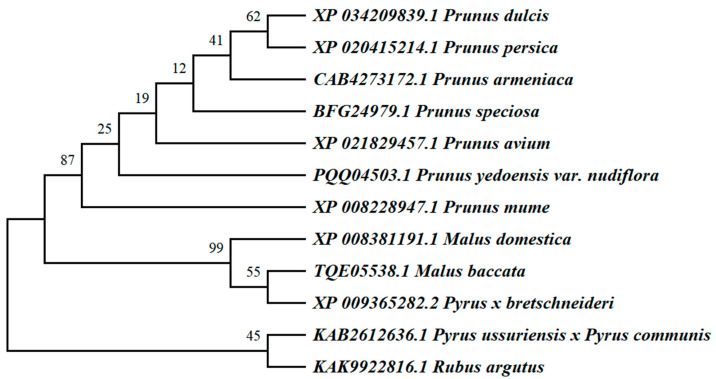
Evolutionary relationship analysis of *WRKY69* protein in *M. domestica* and other species.

**Figure 5 cimb-47-00576-f005:**
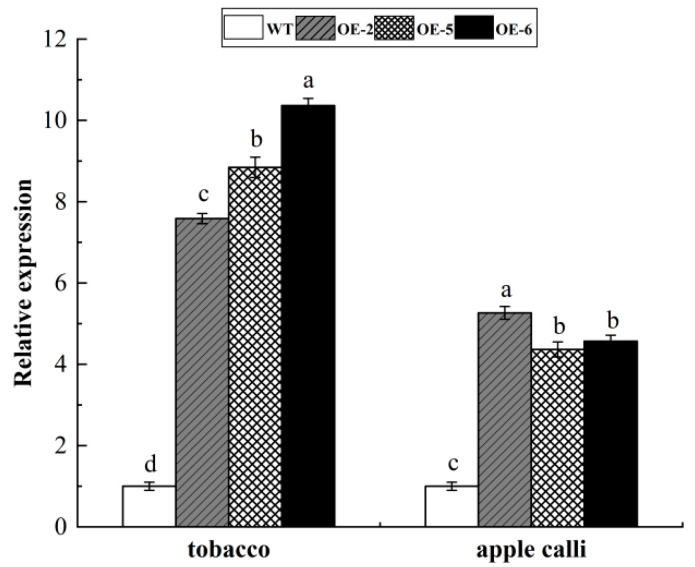
Expression level of WRKY69 TF in three transgenic tobacco and apple calli lines. Note: different letters above the bars indicated significant differences (*p* < 0.05) as assessed by one-way ANOVA and the least significant difference (LSD) test.

**Figure 6 cimb-47-00576-f006:**
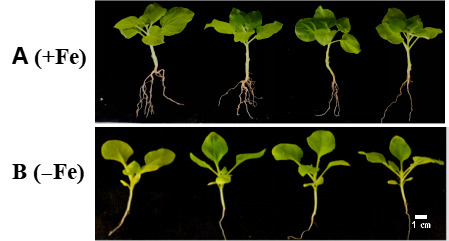
Phenotypes of *WRKY69* transgenic tobacco and WT control tobacco grown for 30 days on Fe-sufficient or Fe-deficient media. (**A**). Fe-sufficient condition. (**B**). Fe-deficient condition.

**Figure 7 cimb-47-00576-f007:**
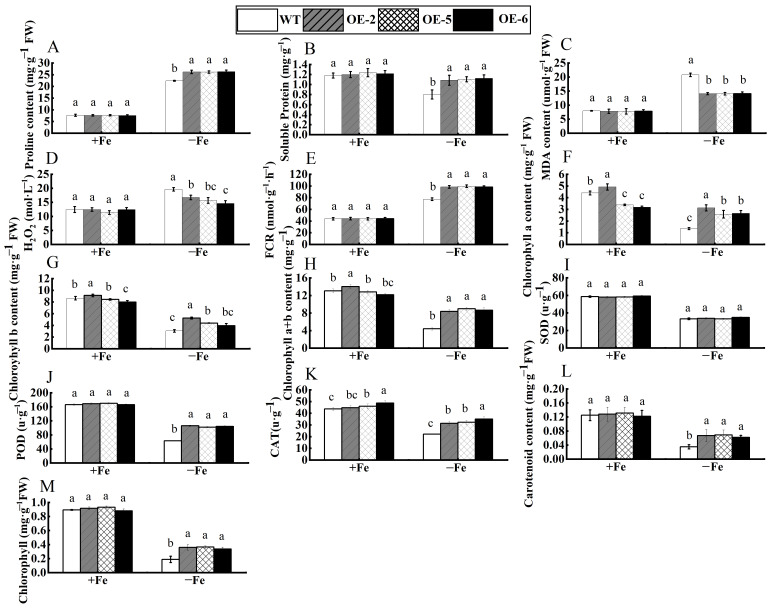
Physiological indices of *WRKY69* transgenic and wild-type (WT) control tobacco grown for 30 days on Fe-sufficient or Fe-deficient media. (**A**). Pro content. (**B**). Soluble protein. (**C**). MDA content. (**D**). H_2_O_2_ content. (**E**). FCR activity. (**F**). Chlorophyll a content. (**G**). Chlorophyll b content. (**H**). Chlorophyll a + b content. (**I**). SOD activity. (**J**). POD activity. (**K**). CAT activity. (**L**) Carotenoid content. (**M**) Chlorophyll content. Note: different letters above the bars indicated significant differences (*p*  <  0.05) as assessed by one-way ANOVA and the least significant difference (LSD) test.

**Figure 8 cimb-47-00576-f008:**
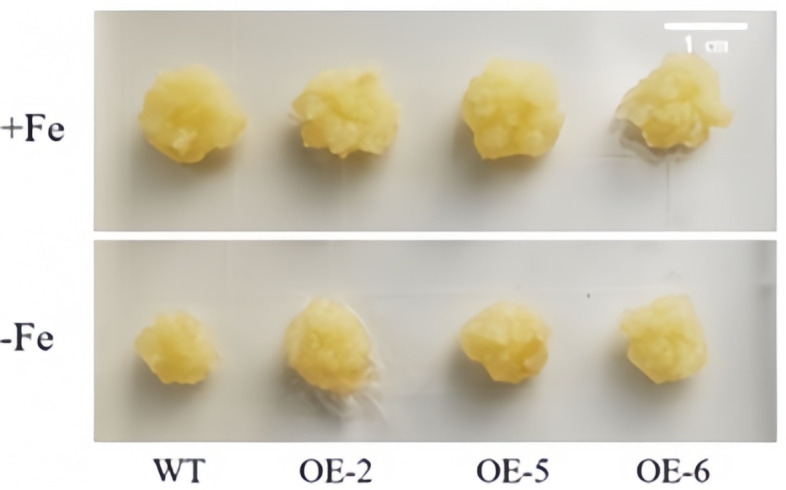
Status of *WRKY69* transgenic and wild-type (WT) control apple calli grown on Fe-sufficient or Fe-deficient media for 20 days.

**Figure 9 cimb-47-00576-f009:**
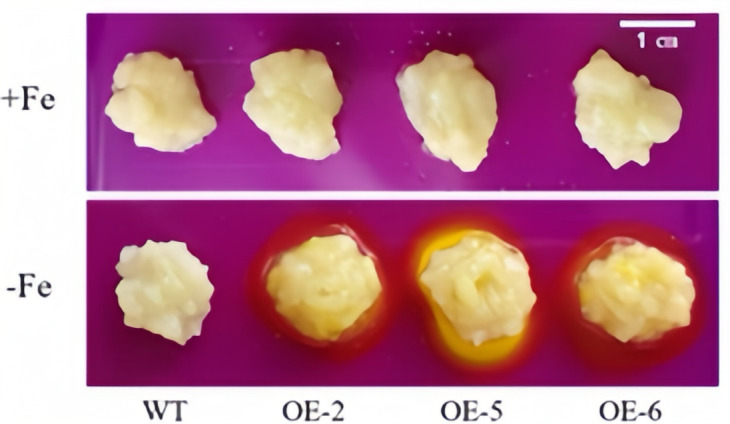
Acidification analysis of transgenic *WRKY69* apple calli on medium containing the pH indicator dye bromocresol violet. A yellow color around apple calli indicates acidification.

**Figure 10 cimb-47-00576-f010:**
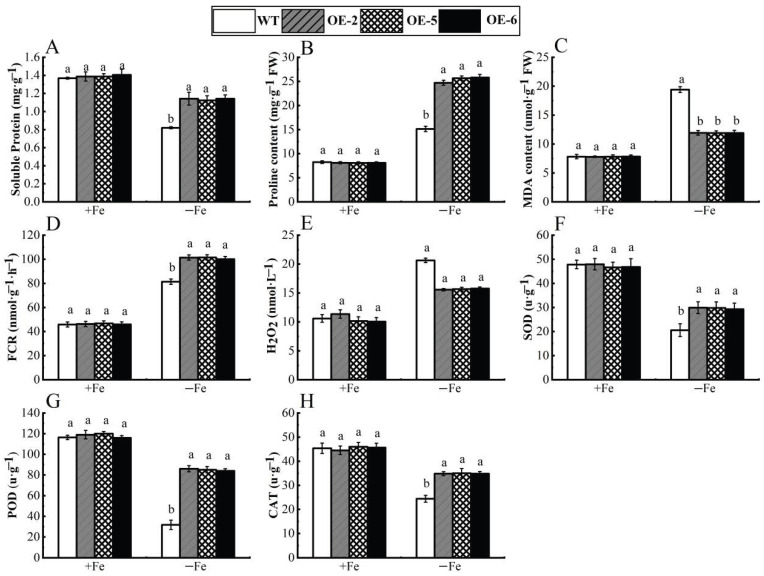
Physiological indices *WRKY69* transgenic and wild-type apple calli grown for 20 days on Fe-sufficient or Fe-deficient media. (**A**). Soluble protein. (**B**). Pro content. (**C**). MDA content. (**D**). FCR activity. (**E**). H_2_O_2_ content. (**F**). SOD activity. (**G**). POD activity. (**H**). CAT activity. Note: different letters above the bars indicated significant differences (*p* < 0.05) as assessed by one-way ANOVA and the least significant difference (LSD) test.

**Table 1 cimb-47-00576-t001:** Physicochemical property analysis of *WRKY69* gene in apple.

Gene ID	Gene Name	Size/aa	Mw /kDa	Instability Index	pI	Aliphatic Index	GRAVY
MD09G1235100	*WRKY69*	250	61.739	45.46	5.10	28.29	0.916

## Data Availability

The data presented in this study is available on request from the corresponding author.

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
