# Peer review of "Functional Analysis of Malus halliana WRKY69 Transcription Factor (TF) Under Iron (Fe) Deficiency Stress"

_cimb, 2025, doi:10.3390/cimb47070576_

Round 1
Reviewer 1 Report
Comments and Suggestions for Authors
This paper reports on the characterization of the MhWRKY69-like gene isolated from Malus halliana by transgenic experiments using tobacco plants and apple callus. The results show that overexpression of the MhWRKY69-like gene by the 35S CaMV promoter (pRI101 vector) conferred tolerance to Fe-deficient stress in tobacco plants and apple calli. This manuscript contains enough data from the experiments. However, several essential explanations about the experiments are not described, and it seems there is a critical error in the data presentation in this version. The data of this study are untrustworthy. I recommend that this manuscript be carefully revised and resubmitted.
- How did the authors find and obtain the WRKY69 homologous genes from the apple genome database? Why is only the MD09G1235100 gene focused? There are many WRKY transcription factor genes in the plant genome, and they constitute a gene family. Notably, Malus species are generally known to be amphidiploid (tetraploid). Therefore, the other genes homologous to WRKY69 possibly exist in the genome. At least two or several homologous genes that have close DNA sequences to WRKY69 should be obtained and analyzed in this study. Why is the sequence of the MD09G1235100 gene not included in Figures 1 and 2? The genome data analysis and gene search seem insufficient.
- Two independent transgenic experiments created transgenic tobacco plants and apple calli, respectively. However, the same-name transgenic lines (transgenic individuals, “OE-2”, “OE-5”, and “OE-6”) were chosen and used in tobacco plants and apple calli to investigate the properties of transgenic lines. This causes a lot of confusion and loses credibility with the data in this paper. Figures 4, 5, 6, 7, 8, and 9 are untrustworthy.
Minor errors
L.13; Real-time quantitative fluorescence >> Real-time quantitative PCR
L.14; WRKY69-like TF expression >> WRKY69-like TF gene expression
L.27; tobacco and apple calli >> tobacco plants and apple calli
L.128; by homologous recombination, >> by a ligase (T4 ligase?)
Which technology was applied to construct a binary vector, In-Fusion (etc.) technology, or a conventional method?
L.166; DAB >> DAB (3,3′-Diaminobenzidine)
L.167; NBT >> NBT(nitro blue tetrazolium)
L.171; MDA >> malondialdehyde (MDA)
L.176; FCR >> Ferric Chelate Reductase (FCR)
L.237; Figures 5A-B, >> Figure 5
L.242; Figures 6 >> Figure 6
L.262; Figure 5 << Add scale bars in the photos
L.264; Fe-deffcient >> Fe-deficient
L.264; Fe-sufffcient >> Fe-sufficient
L.264; Fe-deffcient >> Fe-deficient
L.269; Figure 6 << Add an explanation of error bars in the bar graphs.
L.285; Figure 7 << Add scale bars in the photos
L.288; Figure 8 << Add scale bars in the photos
L.289; Acidiffcation >> Acidification
L.302; WRKY IF >> WRKY TF
L.356; tobacco and apple calli >> tobacco plants and apple calli
L.369; tobacco and apple calli >> tobacco plants and apple calli
L.382-383; tobacco and apple calli >> tobacco plants and apple calli
L.390; tobacco and apple calli >> tobacco plants and apple calli
L.394; tobacco and apple calli >> tobacco plants and apple calli
Author Response
This paper reports on the characterization of the MhWRKY69-like gene isolated from Malus halliana by transgenic experiments using tobacco plants and apple callus. The results show that overexpression of the MhWRKY69-like gene by the 35S CaMV promoter (pRI101 vector) conferred tolerance to Fe-deficient stress in tobacco plants and apple calli. This manuscript contains enough data from the experiments. However, several essential explanations about the experiments are not described, and it seems there is a critical error in the data presentation in this version. The data of this study are untrustworthy. I recommend that this manuscript be carefully revised and resubmitted.
- How did the authors find and obtain the WRKY69 homologous genes from the apple genome database? Why is only the MD09G1235100 gene focused? There are many WRKY transcription factor genes in the plant genome, and they constitute a gene family. Notably, Malus species are generally known to be amphidiploid (tetraploid). Therefore, the other genes homologous to WRKY69 possibly exist in the genome. At least two or several homologous genes that have close DNA sequences to WRKY69 should be obtained and analyzed in this study. Why is the sequence of the MD09G1235100 gene not included in Figures 1 and 2? The genome data analysis and gene search seem insufficient.
Response: Thank you very much for the reviewers' comments. The transcription factor WRKY69 has low homologous protein similarity in apple and no homologous genes. In addition, the sequence of the MD09G1235100 gene is included in Figures 1 and 2, which is the sequence corresponding to malus domestica.
- Two independent transgenic experiments created transgenic tobacco plants and apple calli, respectively. However, the same-name transgenic lines (transgenic individuals, “OE-2”, “OE-5”, and “OE-6”) were chosen and used in tobacco plants and apple calli to investigate the properties of transgenic lines. This causes a lot of confusion and loses credibility with the data in this paper. Figures 4, 5, 6, 7, 8, and 9 are untrustworthy.
Response: Thank you very much for the reviewers' comments. Multiple transgenic lines were obtained for both tobacco and apple calli. Coincidentally, the lines OE-2, OE-5 and OE-6 for tobacco and apple calli were all successfully obtained and grew well. Therefore, they are used for functional verification.
Minor errors
L.13; Real-time quantitative fluorescence >> Real-time quantitative PCR
Response: Thank you very much for the reviewers' comments. “Real-time quantitative fluorescence” has been replaced by “Real-time quantitative PCR”.
L.14; WRKY69-like TF expression >> WRKY69-like TF gene expression
Response: Thank you very much for the reviewers' comments. “WRKY69-like TF expression” has been replaced by “WRKY69-like TF gene expression”.
L.27; tobacco and apple calli >> tobacco plants and apple calli
Response: Thank you very much for the reviewers' comments. “tobacco and apple calli” has been replaced by “tobacco plants and apple calli”.
L.128; by homologous recombination, >> by a ligase (T4 ligase?)
Which technology was applied to construct a binary vector, In-Fusion (etc.) technology, or a conventional method?
Response: Thank you very much for the reviewers' comments. “On the other hand, the pRI101 vector was double-digested with two enzymes (Sma I and Kpn I) for linearization treatment, and then ligated with the recombinant enzyme Exnase II (C112, purchased from Vazyme, Nanjing, China)” has been wrote in “2.3. Cloning and Expression Vector Construction of WRKY69-like Gene” part, and mark it in red.
L.166; DAB >> DAB (3,3′-Diaminobenzidine)
Response: Thank you very much for the reviewers' comments.“DAB” has been replaced by “DAB (3,3′-Diaminobenzidine)”.
L.167; NBT >> NBT(nitro blue tetrazolium)
Response: Thank you very much for the reviewers' comments.“NBT” has been replaced by “NBT(nitro blue tetrazolium)”.
L.171; MDA >> malondialdehyde (MDA)
Response: Thank you very much for the reviewers' comments.“MDA” has been replaced by “malondialdehyde (MDA)”.
L.176; FCR >> Ferric Chelate Reductase (FCR)
Response: Thank you very much for the reviewers' comments.“FCR” has been replaced by “Ferric Chelate Reductase (FCR)”.
L.237; Figures 5A-B, >> Figure 5
Response: Thank you very much for the reviewers' comments.“Figures 5A-B” has been replaced by “Figure 5”.
L.242; Figures 6 >> Figure 6
Response: Thank you very much for the reviewers' comments. “Figures 6” has been replaced by “Figure 6”.
L.262; Figure 5 << Add scale bars in the photos
Response: Thank you very much for the reviewers' comments. Figure 5 has been added scale bars in the photos.
L.264; Fe-deffcient >> Fe-deficient
Response: Thank you very much for the reviewers' comments. “Fe-deffcient” has been replaced by “Fe-deficient”.
L.269; Figure 6 << Add an explanation of error bars in the bar graphs.
Response: Thank you very much for the reviewers' comments. Figure 6 has been added an explanation of error bars in the bar graphs (Note: different letters above the bars indicated significant differences (P < 0.05) as assessed by one-way ANOVA and the least significant difference (LSD) test).
L.285; Figure 7 << Add scale bars in the photos
Response: Thank you very much for the reviewers' comments. Figure 7 has been added scale bars in the photos.
L.288; Figure 8 << Add scale bars in the photos
Response: Thank you very much for the reviewers' comments. Figure 8 has been added scale bars in the photos.
L.289; Acidiffcation >> Acidification
Response: Thank you very much for the reviewers' comments. “Acidiffcation” has been replaced by “Acidification”.
L.302; WRKY IF >> WRKY TF
Response: Thank you very much for the reviewers' comments. “WRKY IF” has been replaced by “WRKY TF”.
L.356; tobacco and apple calli >> tobacco plants and apple calli
Response: Thank you very much for the reviewers' comments. “tobacco and apple calli” has been replaced by “tobacco plants and apple calli”.
L.369; tobacco and apple calli >> tobacco plants and apple calli
L.382-383; tobacco and apple calli >> tobacco plants and apple calli
L.390; tobacco and apple calli >> tobacco plants and apple calli
L.394; tobacco and apple calli >> tobacco plants and apple calli
Response: Thank you very much for the reviewers' comments. “tobacco and apple calli” has been replaced by “tobacco plants and apple calli”. in Lines 356, 369, 382-383, 390 and 394.
Reviewer 2 Report
Comments and Suggestions for Authors
In this manuscript, HongJia Luo and colleagues conducted te functional analysis of Malus halliana MhWRKY69-like transcription factor (TF) under iron (Fe) deficiency stress. I have following comments:
1, For the Abstract, Real-time quantitative fluorescence (qRT-PCR) should be Real-time quantitative fluorescence (RT-qPCR). More values and details should be provided. For instance, authors stated that “the apple calli that were overexpressed WRKY69-like also exhibited superior growth and quality. Furthermore, the overexpression of the WRKY69-like gene enhanced the ability of tobacco to Fe deficiency stress tolerance by stimulating the synthesis of photosynthetic pigments, increasing antioxidant enzyme activity, and facilitating Fe reduction. Additionally, it increased the resistance of apple calli to Fe deficiency stress by enhancing Fe reduction, and elevating the activity of antioxidant enzymes.”, please provide the value. In addition, practical interest of this study should be stated.
2, For the key words, deffciency stress is a mistake.
3, For the introduction, full name of the abbreviations like ALMT1 should be spelt out at their first appearance.
4, For the results, there is no information provided for transformation efficiency and the copy number of transgene in any independent transgenic line. In addition, as a complete story, silencing of WRKY46should be performed.
5, For the materials and methods, genotypes of Malus halliana examined in this study should be described. Biological and technical replicates, as well as randomization methods should be clearly stated. Sampling size should be included.
6, For the discussion, I would like to see the discussion section was divided into subsections with appropriate titles.
7, For the conclusion, practical interest of this study should be included.
Author Response
In this manuscript, HongJia Luo and colleagues conducted te functional analysis of Malus halliana MhWRKY69-like transcription factor (TF) under iron (Fe) deficiency stress. I have following comments:
1, For the Abstract, Real-time quantitative fluorescence (qRT-PCR) should be Real-time quantitative fluorescence (RT-qPCR). More values and details should be provided. For instance, authors stated that “the apple calli that were overexpressed WRKY69-like also exhibited superior growth and quality. Furthermore, the overexpression of the WRKY69-like gene enhanced the ability of tobacco to Fe deficiency stress tolerance by stimulating the synthesis of photosynthetic pigments, increasing antioxidant enzyme activity, and facilitating Fe reduction. Additionally, it increased the resistance of apple calli to Fe deficiency stress by enhancing Fe reduction, and elevating the activity of antioxidant enzymes.”, please provide the va lue. In addition, practical interest of this study should be stated.
Response: Thank you very much for the reviewers' comments. “Real-time quantitative fluorescence (qRT-PCR)” has been replaced by “Real-time quantitative fluorescence (RT-qPCR)”. The value also has been provide. At last, practical interest of this study also has been added.
2, For the key words, deffciency stress is a mistake.
Response: Thank you very much for the reviewers' comments. “deffciency stress” has been replaced by “deficiency stress”.
3, For the introduction, full name of the abbreviations like ALMT1 should be spelt out at their first appearance.
4, For the results, there is no information provided for transformation efficiency and the copy number of transgene in any independent transgenic line. In addition, as a complete story, silencing of WRKY46 should be performed.
Response: Thank you very much for the reviewers' comments. As a complete story, WRKY46 should be silenced. In the future, we will definitely follow your advice and silenced this gene. Due to time constraints this time, I was unable to make up for this part of the experiment. We apologize for any inconvenience this may cause.
5, For the materials and methods, genotypes of Malus halliana examined in this study should be described. Biological and technical replicates, as well as randomization methods should be clearly stated. Sampling size should be included.
Response: Thank you very much for the reviewers' comments. Relevant information of Malus halliana examined in this study should be described. In addition, “Biological and technical replicates, as well as randomization methods should be clearly stated. Sampling size should be included.” has been added in materials and methods part, and marked in red.
6, For the discussion, I would like to see the discussion section was divided into subsections with appropriate titles.
Response: Thank you very much for the reviewers' comments. The discussion section has been divided into subsections with appropriate titles (4.1 Bioinformatics analysis of MhWRKY69-like TF; 4.2 Overexpression of MhWRKY69-like enhanced tolerance to Fe deficiency in transgenic plants and apple calli).
7, For the conclusion, practical interest of this study should be included.
Response: Thank you very much for the reviewers' comments. The practical interest of this study has been added (In conclusion, the screening and identification of WRKY family members in apple that respond to Fe deficiency stress, not only enrich the network of WRKY family members responding to Fe deficiency stress, but also contribute to the innovation of apple stress-resistant germplasm resources, which holds significant scientific significance and application prospects).
Reviewer 3 Report
Comments and Suggestions for Authors
- Add the significance of this study and some future recommendations in the conclusion.
- Make all gene names italic, especially in references.
- Why this gene was chosen is not mentioned anywhere in the Materials and Methods section.
- Check section 2.6 title "Determination should be in small letter also, Fe both abbreviation and the full form are mentioned many times, it should be only one time (the first time).
- Line 172, write full form of "SOD, POD, and CAT during first time.
- Line 163. give space to the last sentence at start.
- Line 157, and 161 too many repeatation of Fe with both abbreavtion anf full form.
- Lin 169, add more detail about chlorohyll content how it was measure and why only chl-a/b, and why Caro is not included. I suggest adding total chlorphyll too.
- The manuscript is fine, but a serious and carefully look and need revise the language as well. The current version has many small errors and rough language. I suggest checking the language by a native English speaker.
The English language needs to be revised.
Author Response
Add the significance of this study and some future recommendations in the conclusion.
Response: Thank you very much for the reviewers' comments. The significance of this study and some future recommendations in the conclusion has been added (In conclusion, the screening and identification of WRKY family members in apple that respond to Fe deficiency stress, not only enrich the network of WRKY family members responding to Fe deficiency stress, but also contribute to the innovation of apple stress-resistant germplasm resources, which holds significant scientific significance and application prospects. In the future, a variety of biochemical techniques will be utilized, such as electrophoretic mobility assay (EMSA), to search for the downstream target genes of the WRKY69-like TF, and experiments such as yeast double heterozygotic (Y2H), and co-immunoprecipitation (Co-IP) will be used to search for the interacting proteins of this transcription factor, which reveal the molecular mechanism by which the TF regulates Fe deficiency stress.).
Make all gene names italic, especially in references.
Response: Thank you very much for the reviewers' comments. All the gene names have been italicized.
Why this gene was chosen is not mentioned anywhere in the Materials and Methods section.
Response: Thank you very much for the reviewers' comments. Why this gene was chosen has been mentioned in the last paragraph of the introduction and has been marked in red.
Check section 2.6 title "Determination should be in small letter also, Fe both abbreviation and the full form are mentioned many times, it should be only one time (the first time).
Response: Thank you very much for the reviewers' comments. “Determination” has been iin lowercase.
Line 172, write full form of "SOD, POD, and CAT during first time.
Response: Thank you very much for the reviewers' comments. The full form of "SOD, POD, and CAT during first time have been wrote ( superoxide dismutase (SOD), peroxidase (POD) and catalase (CAT) activities).
Line 163. give space to the last sentence at start.
Response: Thank you very much for the reviewers' comments. The space to the last sentence at start has been added.
Line 157, and 161 too many repeatation of Fe with both abbreavtion anf full form.
Response: Thank you very much for the reviewers' comments. The repetition of the abbreviation form and the full name form of Fe has been streamlined.
Lin 169, add more detail about chlorohyll content how it was measure and why only chl-a/b, and why Caro is not included. I suggest adding total chlorphyll too.
Response: Thank you very much for the reviewers' comments. “Caro” and “chlorphyll content” has been supplemented in Figure 6.
The manuscript is fine, but a serious and carefully look and need revise the language as well. The current version has many small errors and rough language. I suggest checking the language by a native English speaker.
Response: Thank you very much for the reviewers' comments. The language of the entire manuscript has been checked and revised, and marked in red. For example, “The WRKY69-like gene (MD09G1235100), isolated from Malus halliana, features an open reading frame of 753 bp. ” has been replaced by “The WRKY69-like gene (MD09G1235100) was isolated from Malus halliana, with an open reading frame of 753 bp. ”.
Round 2
Reviewer 1 Report
Comments and Suggestions for Authors
The screening process of the WRKY69-like gene was not well described. I’m wondering why only the WRKY69-like gene is focused on from the WRKY gene family. The analysis and information on homologous genes of WRLY69 in Malus halliana are necessary in this paper.
I conducted a BLAST search using a sequence of the XP_008381191 gene described in this manuscript. I noticed that many of its homologues, including WRKY98, WRKY91, WRKY15, WRKY65, WRKY3, and WRKY14, are registered in the NABI (GenBank) database, even in Malus species.
The data of sequence analysis, phylogenetic analysis, and expression analysis of multiple WRKY homologous genes should be included in this paper.
Author Response
The screening process of the WRKY69-like gene was not well described. I’m wondering why only the WRKY69-like gene is focused on from the WRKY gene family.
Response: Thank you very much for the reviewers' comments. The screening process of the WRKY69-like gene has been well described in Introduction Part (Previously, transcriptomic analysis revealed that WRKY69 was a differentially expressed TF under Fe deficiency conditions. This finding was further validated by RT-qPCR, which confirmed that WRKY69 expression was significantly upregulated in response to Fe deficiency. Consequently, this TF was cloned for subsequent genetic transformation and functional characterization).
The analysis and information on homologous genes of WRLY69 in Malus halliana are necessary in this paper.
I conducted a BLAST search using a sequence of the XP_008381191 gene described in this manuscript. I noticed that many of its homologues, including WRKY98, WRKY91, WRKY15, WRKY65, WRKY3, and WRKY14, are registered in the NABI (GenBank) database, even in Malus species.
The data of sequence analysis, phylogenetic analysis, and expression analysis of multiple WRKY homologous genes should be included in this paper.
Response: Thank you very much for the reviewers' comments. The analysis of the homologous genes of WRKY69 and the analysis of related information are very necessary. However, through NCBI blast and GDR blast, this transcription factor has no homologous genes in apple. The results are shown in the following figure.
Reviewer 2 Report
Comments and Suggestions for Authors
Authors have addressed my concerns in the revision.
Author Response
Thank you very much for the recognition of the reviewers
Round 3
Reviewer 1 Report
Comments and Suggestions for Authors
The WRKY gene family in Malus domestica has been previously analyzed and reported.
Qin, Y.; Yu, H.; Cheng, S.; Liu, Z.; Yu, C.; Zhang, X.; Su, X.; Huang, J.; Shi, S.; Zou, Y.; et al. Genome-Wide Analysis of the WRKY Gene Family in Malus domestica and the Role of MdWRKY70L in Response to Drought and Salt Stresses. Genes 2022, 13, 1068. https://doi.org/10.3390/genes13061068
This paper is not referred to in this manuscript.
In the previous paper (Qin et al., 2022), shows that there are two homologs close to the apple WRKY69 gene (MD09G1235100).
Those are MD12G1243400 and MD04G1226400.
Moreover, WRKY11 and WRKY17 are also located near WRKY69.
The protein sequence of the apple WRKY69 is available in the public database, accession "XP_008381191". That sequence is as follows.
>XP_008381191.1 probable WRKY transcription factor 69 [Malus domestica]
MEDSPSSASEDDTQVATPLPKKRRGVQKRVVSVPIGDVEGSKSKGEGHPPSDSWAWRKYGQKPIKGSPYP
RGYYRCSSSKGCPARKQVERSRLNPTMLIVTYVCDHNHPMPTTKANQSSSVTPTTITATATSPTTESELP
AKTEELTIFTSQVDLDLSDDSPTLLSDFGWFNDAASTVVLESPICLGNSSCMNNDVATRLKDEDEYLFAD
LGELPEGSVIFRHRNRMVESDDQNRRCSLSAVVPYCSNTG
The BLAST search using this sequence results in the display of many homologous genes from the genome database of Malus species.
The below response comment from the authors is incorrect and unbelievable.
"Response: Thank you very much for the reviewers' comments. The analysis of the homologous genes of WRKY69 and the analysis of related information are very necessary. However, through NCBI blast and GDR blast, this transcription factor has no homologous genes in apple. The results are shown in the following figure. "
Author Response
- The WRKY gene family in Malus domestica has been previously analyzed and reported.
Qin, Y.; Yu, H.; Cheng, S.; Liu, Z.; Yu, C.; Zhang, X.; Su, X.; Huang, J.; Shi, S.; Zou, Y.; et al. Genome-Wide Analysis of the WRKY Gene Family in Malus domestica and the Role of MdWRKY70L in Response to Drought and Salt Stresses. Genes 2022, 13, 1068. https://doi.org/10.3390/genes13061068
This paper is not referred to in this manuscript.
Response: Thank you very much for the reviewers' comments. This manuscript has cited this paper in the discussion section, and the reference number is 51.
- In the previous paper (Qin et al., 2022), shows that there are two homologs close to the apple WRKY69 gene (MD09G1235100).
Those are MD12G1243400 and MD04G1226400.
Moreover, WRKY11 and WRKY17 are also located near WRKY69.
The protein sequence of the apple WRKY69 is available in the public database, accession "XP_008381191". That sequence is as follows.
>XP_008381191.1 probable WRKY transcription factor 69 [Malus domestica]
MEDSPSSASEDDTQVATPLPKKRRGVQKRVVSVPIGDVEGSKSKGEGHPPSDSWAWRKYGQKPIKGSPYP
RGYYRCSSSKGCPARKQVERSRLNPTMLIVTYVCDHNHPMPTTKANQSSSVTPTTITATATSPTTESELP
AKTEELTIFTSQVDLDLSDDSPTLLSDFGWFNDAASTVVLESPICLGNSSCMNNDVATRLKDEDEYLFAD
LGELPEGSVIFRHRNRMVESDDQNRRCSLSAVVPYCSNTG
The BLAST search using this sequence results in the display of many homologous genes from the genome database of Malus species.
Response: Thank you very much for the reviewers' comments. The homologous gene of WRKY69 has been identified and evolutionary analysis, multiple sequence alignment, and fluorescence quantitative analysis under Fe deficiency stress have been conducted, and supplement it in Section 3.1. Protein sequence analysis and evolutionary relationship analysis of WRKY69 and its homologous genes in 3 results section .